# Elucidating the Role of Santalol as a Potent Inhibitor of Tyrosinase: In Vitro and In Silico Approaches

**DOI:** 10.3390/molecules27248915

**Published:** 2022-12-15

**Authors:** Nabeel Ali, Zainy Zehra, Anas Shamsi, Md. Amjad Beg, Zahoor Ahmad Parray, Md. Ali Imam, Naseem A. Gaur, Md. Imtaiyaz Hassan, Anis Ahmad Chaudhary, Hassan Ahmad Rudayni, Mohammed Ibrahim Alghonaim, Sulaiman A. Alsalamah, Asimul Islam

**Affiliations:** 1Centre for Interdisciplinary Research in Basic Sciences, Jamia Millia Islamia, New Delhi 110025, India; 2International Centre for Genetic Engineering and Biotechnology, Aruna Asaf Ali Marg, New Delhi 110067, India; 3Department of Biology, College of Science, Imam Mohammad Ibn Saud Islamic University (IMSIU), Riyadh 11623, Saudi Arabia

**Keywords:** mushroom tyrosinase, melanogenesis, tyrosinase inhibitor, multi-spectroscopic techniques, natural compounds, tyrosinase inhibitor

## Abstract

This research work focuses on the potential application of an organic compound, santalol, obtained from santalum album, in the inhibition of the enzyme tyrosinase, which is actively involved in the biosynthesis of melanin pigment. Over-production of melanin causes undesirable pigmentation in humans as well as other organisms and significantly downgrades their aesthetic value. The study is designed to explain the purification of tyrosinase from the mushroom *Agaricus bisporus*, followed by activity assays and enzyme kinetics to give insight into the santalol-modulated tyrosinase inhibition in a dose-dependent manner. The multi-spectroscopic techniques such as UV-vis, fluorescence, and isothermal calorimetry are employed to deduce the efficiency of santalol as a potential candidate against tyrosinase enzyme activity. Experimental results are further verified by molecular docking. Santalol, derived from the essential oils of santalum album, has been widely used as a remedy for skin disorders and a potion for a fair complexion since ancient times. Based on enzyme kinetics and biophysical characterization, this is the first scientific evidence where santalol inhibits tyrosinase, and santalol may be employed in the agriculture, food, and cosmetic industries to prevent excess melanin formation or browning.

## 1. Introduction

Melanin is a light-absorbing bio pigment with coloration ranging from dark blackish-brown eumelanin to reddish-yellow pheomelanin [1]. This molecule imparts a specific colorway to the skin, eyes, and hair in humans [2]. Melanin is also synthesized in various fruits and vegetables, which can be seen as a colored pigment. Melanin shields the skin from ultraviolet (UV) rays in the sunlight by absorbing the harmful radiation [3]. Thus, melanin production is an important process; by contrast, there are occurrences where enhanced melanin production is a serious problem [4]. Skin disorders such as freckles, lentigo, melasma, or pigmented acne seen across all races are the result of an uncontrollable accumulation of melanin [5,6]. These skin disorders not only affect the individual’s physical appearance, but they may elicit social and mental insecurities resulting in lower quality of life (QOL) [7,8]. Another grave concern is the process of neuromelanin occurring in the brain, where dopamine is oxidized to dopaquinones by the action of tyrosinase [9]. Previous studies revealed that hyper tyrosinase activity in dopamine-rich nigral neurons can lead to neuronal tissue damage and apoptotic cell death. Neuromelanin is often associated with neurodegenerative diseases like Parkinson’s disease and Huntington’s disease [10]. Similarly, prolonged melanin synthesis develops brownish-black melanin spots over food stocks such as fruits and vegetables and subsequently leads to a significant drop in their nutritional as well as market values, accounting for huge economic loss [11]. This biosynthetic reaction of melanin synthesis takes place in melanocytes where the amino acid L- tyrosine is converted to the end-product melanin in a series of steps with the participation of a key enzyme called tyrosinase [12,13]. This enzyme catalyzes two fundamental steps of melanin production, which begin with the hydroxylation of monophenols to *o*-diphenols, namely L-DOPA (L-3,4-dihydorxyphenylalanine) displaying its monophenolase activity. The second step is the oxidation of *o*-diphenols to *o*-quinones, namely dopaquinone presenting its diphenolase activity. Subsequently, *o*-quinones polymerize to produce melanin in a cascade of reactions [14] (Figure 1). Evidently, tyrosinase is a polyphenoloxidase (EC 1.14.18.1), regulatory metallo-enzyme having two copper ions at its active site, with several isoforms. The enzyme is reported to have approximately 120 kDa of molecular mass, distributed among two H subunits, each of ∼43 kDa, and two L subunits, each of ∼14 kDa, which is the most common form of tyrosinase. Thus, existing as an H_2_L_2_ tetramer in the holoenzyme state [15,16], it possesses two binding sites for the phenolic substrates along with a copper-conjugated oxygen binding site. The fact that mushroom tyrosinase obtained from *Agaricus bisporus* has striking homology with mammalian tyrosinase [17] therefore makes it a convenient target for inhibition studies for therapeutics applications in humans as well as in the food industries as food preservatives [18].

As stated above, tyrosinase acts as a rate-limiting enzyme in the melanin biosynthesis pathway. Thus, the primary approach to slow down the pathway is to inhibit tyrosinase via different strategies, of which anti-tyrosinase action by chemical modalities, such as kojic acid, arbutin, and hydroquinones, has been widely used [20,21]. Although these compounds with both natural and synthetic origins have been reported to possess tyrosinase inhibitory properties, they are often limited subject to their toxicity, activity, or stability [20,22,23]. Thus, a naturally occurring compound with an inhibitory effect on melanogenesis serves as a good choice for commercial applications because of its organic nature and accompanied by less toxicity and the highest compatibility when compared to its synthetic counterparts [24]. Therefore, we screened a phytochemical called santalol, derived from the essential oils of sandalwood (*santalum album*). In traditional medicinal systems, it is widely used as a remedy for skin disorders and a potion for a fair complexion [25]. Chemically, santalol is a phenolic compound with good antioxidant abilities [26]. In addition, it is approved by the FDA for human use and consumption [27,28]. Apparently, there is a lack of information on the pharmacological potency of santalol against tyrosinase activity [29]. Here, in this study, we have investigated its role as a tyrosinase inhibitor along with a plausible mechanism of inhibition by biophysical study. Furthermore, using multi-spectroscopic techniques, we have elucidated the interaction mechanism of the pure santalol compound towards the purified enzyme tyrosinase, later confirmed by molecular docking results.

## 2. Results and Discussion

### 2.1. Extraction of Enzyme Tyrosinase and Ammonium Sulphate Precipitation

A cell-free mushroom tyrosinase system was used to conveniently extract the target protein. A total of 310 mL crude protein extract was used from the 200 g of mushroom taken as the source. The ammonium sulphate precipitation method was used to precipitate the crude protein by saturating the crude solution with finely divided ammonium sulphate in a stepwise manner. The concept of salting-out was applied, where higher salt concentrations decrease the protein solubility in a solution leading to precipitation of the protein, which eventually comes out of the solution and can easily be recovered after centrifugation [30,31]; we obtained most of the protein, including the target protein (protein of interest, tyrosinase) in a 60% fraction of ammonium sulphate salt concentration. 

### 2.2. Purification of Enzyme Tyrosinase Using Chromatography

The target protein is often accompanied by other proteins and organic molecules which tend to accumulate during homogenized extract preparation. The two-step purification was performed to segregate the target protein from the mixture using ion-exchange and gel filtration chromatography. When dialyzed protein sample was applied to a DEAE-cellulose column with stepwise increment in the NaCl gradient, the target protein was eluted at 0–100 mM NaCl with phosphate buffer of the same 6.8 pH. (The result is illustrated in Appendix A). DEAE is basically a positively charged resin that binds to or exchanges opposite-charged (negatively charged) protein molecules present in the protein mixture at a particular operating pH. The eluted active fractions having maximum absorbance at 280 nm were mixed, concentrated, and then loaded to a Superdex 200 pg gel filtration column in order to obtain the purest form of tyrosinase [32,33,34].

Each of the peaks obtained after ion-exchange chromatography was collected as a separate aliquot and run on SDS-PAGE. The purified protein obtained after both chromatographic techniques showed a single band of 63 kDa, as shown in Figure 2. The final eluted fraction showed major peaks of tyrosinase activity which were observed in active fractions with a final specific activity. SDS-PAGE of the enzyme after two step-purification showed that the resolved electrophoretic bands were progressively improved from the crude extract to the final step of the gel filtration column chromatography. It revealed only a single distinctive protein band for the pure form of tyrosinase with an apparent molecular weight of 63 kDa obtained. Total amount and concentration of protein so obtained after every step of purification was estimated with BSA as a standard, and percentage yield is summarized in Table 1.

### 2.3. Tyrosinase Activity Assay

UV-vis absorbance spectral analysis is a widely used methodology to explore the conformational changes occurring in the protein with respect to the formation of protein–ligand complexes. The purified protein so obtained after two-step purification was first subjected to its activity determination using UV-visible spectroscopy and zymogram analysis. The initial rate of reaction is proportional to the concentration of the enzyme. One unit of enzyme corresponded to the amount which catalyzed the transformation of 1 μmol of substrate to product per min under the above conditions and produced changes in absorbance. It has been observed that dopachrome gives maximum absorption at 473 nm [36]. Therefore, we monitored the change in absorption until the horizontal line or point of saturation of the enzyme was achieved; in other words, purified tyrosinase activity was monitored until no L-DOPA (substrate) remained for the enzyme to be converted into the product (dopachrome). As tyrosinase protein activity is based upon the change of color of the L-DOPA substrate to red pigment dopachrome (Figure 3), the intensity of the color is directly proportional to tyrosinase concentration. The zymogram obtained after native-PAGE of the purified protein is depicted in Figure 3 below. Formation of melanin was also confirmed by incubating the native-PAGE for zymography for about 24 h (Appendix A). All these observations clearly indicated that the tyrosinase enzyme that we purified in the end was not only in its purest form (single protein band 63 kDa) but also in its native confirmation (as shown in colorimetric assay and zymography below). 

### 2.4. Tyrosinase Inhibition Kinetics

Changes in the tertiary structure of tyrosinase were monitored using UV-visible scan at 340–240 nm. We observed that, with an increase in santalol concentration from 0 µM to 50 µM, there was perturbation in the tertiary structure of tyrosinase, which was later confirmed by plotting absorption maxima at 280 nm (A_280 nm_) versus santalol concentration (Figure 4A) To elucidate the activity of tyrosinase obtained after purification, an enzyme kinetics study was performed in the absence as well as in the presence of santalol as a tyrosinase inhibitor. Linearized data obtained from change in absorbance per unit time and was fitted by Michaelis–Menten equation using GraphPad prism and sigma plot software version 12.5. The enzyme kinetics, as measured by the Michaelis constant (*K*_m_), are defined as the substrate concentration at which half of the maximum velocity or the rate of the enzymatic reactions achieved. The Michaelis–Menten constant (*K*_m_) value of the purified tyrosinase was estimated in a given range of substrate concentrations for L-dopa (Appendix A). In order to calculate exact *K*_m_ values with different concentrations of inhibitor, a Lineweaver–Burk plot was drawn. The apparent *K*_m_ value of protein was calculated from the Lineweaver–Burk plots relating 1/V to 1/[S]. We observed an overall decrease in enzyme activity of tyrosinase with increase in the concentration of santalol as inhibitor. Maximal inhibition was found to be around 50 µM of santalol.

Using the linearized data obtained by Michaelis–Menten plot, the inhibitory type was also determined by the Lineweaver–Burk plot (Figure 4B); for simplicity, four major concentrations are shown in the plot. Plots clearly show a decrease in overall activity of the purified tyrosinase with increases in santalol concentration (in a dose-dependent manner). Santalol might be bound to the active site of tyrosinase in a competitive manner such that its substrate (L-dopa, used in this study) is no longer accessible to the enzyme for binding, thus its final products, eumelanin or dopachrome, are not formed. Furthermore, on the basis of calorimetric assays, the inhibition of tyrosinase by santalol (extent of inhibition) was confirmed using UV-visible spectroscopic measurement at 473 nm (Appendix A), where we compared the maximal inhibitory concentration of santalol with the standard inhibitor of tyrosinase (kojic acid).

### 2.5. Intrinsic Fluorescence Binding Study

An intrinsic fluorescence study was carried out to examine interactions between protein and the ligand. This can tell us various parameters for drug-protein binding. The intrinsic protein fluorescence is mainly due to aromatic residues such as tyrosine, tryptophan, and phenylalanine in the protein sequence where they act as sensitive fluorescent probes to investigate protein interactions with particular ligands. In addition, strong binding affinities exist between santalol and tyrosinase, possibly due to the presence of hydroxyl groups in santalol. From the result of the intrinsic fluorescence spectra, we observed that santalol binding to the tyrosinase leads to conformational changes in tyrosinase that result in gradual decreases in the spectra in a dose-dependent manner (Figure 5A); for simplicity, we have not shown the overlapping spectra here. Although the decreases in the fluorescence intensities were caused by quenching, there was no significant wavelength shift, indicating that tyrosinase does not undergo unfolding or denaturation after binding with the compound santalol.

To identify the interaction mechanism of the ligand santalol with the enzyme tyrosinase, the data of fluorescence quenching were presented in the form of a Stern–Volmer plot, using the following equation:

Fluorescence quenching was described by the Stern–Volmer equation.
(1)$$\frac{F_{0}}{F}=1+K_{SV}\;\left[C\right]$$
where *F_0_* = fluorescence intensities before the addition of the quencher; *F* = fluorescence intensities after the addition of the quencher; and *K_SV_* = Stern–Volmer quenching constant.

The value of *K*_sv_ determined by the linear regression plot of F_0_/F vs. [Q] at 25 °C was 5.32 × 10^4^. The plot in the Figure 5B showed a good linear relationship using Stern-Volmer (Figure 5B(i)) and modified Stern Volmer (Figure 5B(ii)), indicating that a single type of quenching process (static or dynamic quenching) occurred during the formation of the santalol–tyrosinase complex. For static quenching interactions, if there are similar and independent sites in biological molecules, the apparent binding constants (*K_a_*) and the number of binding sites (*n*) can be obtained from the following equation:log (*F*_0_ − *F*)/*F* = log *K_a_* + nlog Q1⁄2 (2)

According to the intercept and slope value of the regression curve (Figure 5B), the values of *K*_a_ and n for the ligand–tyrosinase interaction were calculated based on Equation (3). The *K*_a_ value of 1.64 × 10^5^ L/mol achieved the order of magnitude of 10^5^ L/mol, indicating the strong binding of the ligand to the binding pocket of enzyme tyrosinase. Moreover, the value of *n* was close to one, suggesting that there was a single binding site or a single class of binding sites in tyrosinase for the given ligand.

### 2.6. Isothermal Titration Calorimetry (ITC)

Isothermal titration calorimetry (ITC) measurements were taken to know the binding affinity of santalol with the purified tyrosinase. ITC is a widely used technique to deduce the interactions between proteins and other molecules based on changes in energy when the two moieties or molecules bind to one another. The baseline was also run with the corresponding buffer for precise measurements and was later subtracted from the main data to obtain the best fitted final figure using various models of sequential binding. Figure 6 depicts graphical outcomes of titrated santalol (ligand) with the protein tyrosinase. The top panel in the figure gives the raw data in power versus time (heat per unit of time liberated from every injection of the ligand with respect to the protein), while the lower panel in the figure displays the raw data in power standardized to the amount of the injectant (kcal mol^−1^) versus its molar ratio of santalol injections into the cell containing tyrosinase. From the data, the thermodynamic binding parameters were calculated and can be seen to show the change in enthalpy (Δ*H*), the association constant (*K*_a_), and the equilibrium dissociation constant, and the change in free energy (Δ*G*°), which was estimated using the equation given below. From Table 2, it can be observed that Δ*G*° is negative and signifies the bi-molecular reaction is spontaneous, and the negative enthalpy change signifies the process is exothermic in nature.

### 2.7. Molecular Docking Analysis

The crystal structure of PPO3 (a tyrosinase enzyme) has two molecules which have tetramer chains. The first is polyphenol oxidase, with tetramers A, B, C, and D chains having 2–392 residues, and the second, lectin-like fold protein, with tetramers E, F, G, H chains having 9–150 residues. In order to acquire more in-depth information on the competitive inhibition mechanism between santalol and PPO3 (tyrosinase) and using kojic acid as standard, molecular docking studies were performed by using InstaDock [37]. In molecular docking, the compounds produced log files, where the affinity (kcal/mol) obtained and docked poses for the discrete compounds analyzed. The experiential binding free energy for the santalol–PPO3 tyrosinase complex was −5.802 kcal mol^−1^ (−24.283 k J mol^−1^), and the standard kojic acid complex was −5.408 mol^−1^ (−22.608 k J mol^−1^). The binding free energy calculations show that santalol binds to PPO3 tyrosinase protein more firmly than kojic acid and the detailed of energies calculation in mentioned in Appendix A. To discover the active sites of tyrosinase, look for neighboring Cu atom residues that are involved in diverse catalytic activities, which are HIS61, HIS85, PHE90, HIS94, HIS259, HIS263, PHE292, HIS295, and HIS296 and are shown in Figure 7A,B. In previous studies, the cavity of tyrosinase comprises two sites: the peripheral and the active site; therefore, it can be seen in Figure 7B that copper, as presented in the yellow sphere in Figure 8B, is brown in color, and the docking studies determined that the Cu B site has the most conspicuous binding in comparison to Cu A, which means it was more tightly bound to the hydroxyl group.

Molecular interaction studies of santalol–PPO3 tyrosinase complex revealed it is closely bound to the true conformation active sites and is most likely contained on the catalytic sites, which is shown in Figure 8A–C. As shown in Figure 8A, the 3D surface representation of the santalol–tyrosinase complex clearly shows the cavity of the binding pocket, whereas Figure 8B shows the cartoon representation of the tyrosinase protein complex with santalol and their interactions are clearly visible. However, in Figure 8C there is a 2D interaction diagram which probes several interacting residues in expanded form for HIS61, HIS244, VAL248, HIS259, ASN260, HIS263, PHE264, MET280, SER282, VAL283, and ALA286, which were involved in hydrogen bonding and hydrophobic interactions. The hydroxyl group of santalol was oriented towards copper (Cu) B, and it was near to MET280 and SER282, which formed a hydrogen bond with 2.26 and 3.35 Å distance.

As far the label of santalol being “safe to use” is concerned [38], toxicity evaluation of santalol by machine learning methods using a pkCSM server [39] were performed (Appendix A). All the predicted outcomes obtained fell under the category of safe to use, which correlates with the origin of the natural source, i.e., sandalwood oil.

The thermodynamic binding parameters and number of binding sites were measured using ITC, whereas molecular docking in support was carried out to know the binding sites of the ligand on the protein and the amino acids involved in the bi-molecular reaction. Coincidently, the Δ*G*° value obtained from the average value of each step of binding in ITC was (−5.946 kcal/mol), which is very close to the one obtained after docking analysis (−5.8 kcal·mol^−1^).

## 3. Materials and Methods

### 3.1. Materials

The target enzyme, tyrosinase was obtained from an edible mushroom (*Agaricus bisporus*). The test compounds, santalol (CAS no. 77-42-9, 93% pure), dimethyl-sulfoxide (DMSO), and 3, 4-Dihydroxyphenylalanine (L-DOPA), were bought from Sigma-Aldrich. All other reagents such as phosphate buffer, Tris buffer, sodium chloride, SDS, etc. of analytical grades were purchased from Merck ltd (Bengaluru, India). 

### 3.2. Methods

#### 3.2.1. Extraction of the Enzyme Tyrosinase

The protocol of Haghbeen et al., with few alterations, was carried out for extraction of mushroom tyrosinase [40]. The mushroom tissues were made into a paste with a blender. The mushroom slurry so obtained was mixed with pre-prepared 200 mL of cold 50 mM phosphate buffer (pH 6.8) using a proportion of 180–200 g of mushroom caps. To remove various undesirable cellular products such as cellulose and cell debris, centrifugation of homogenate for 30 min at 4200× *g* was performed to collect supernatant to be used as a source of enzyme. The pellet obtained after centrifugation was again mixed with cold phosphate buffer, left undisturbed on ice with intermittent shaking, and followed by another round of centrifugation to yield more enzyme from the supernatant. 

#### 3.2.2. Ammonium Sulphate Precipitation with Dialysis

Ammonium sulphate precipitation was done by weighing the finely grounded ammonium sulphate and then mixed in the extract and keep stirring. The mix was centrifuged at 8000× *g* for 20 min at 4 °C. Different degrees of precipitation were tried for enzyme tyrosinase enzyme, i.e., 30%, 60%, 90%, and different precipitates were collected. Dialysis was conducted for different precipitates against 50 mM potassium phosphate buffer (pH 6.8) continuously for 24 h and the buffer was changed periodically after every 7–8 h [41]. The fraction of protein obtained after last dialysis was chosen for the study [42,43].

#### 3.2.3. Purification of Enzyme Tyrosinase Using Chromatography

The purification of crude enzyme obtained after dialysis was exposed to a couple of chromatographic techniques viz. ion-exchange chromatography and gel filtration to obtain the protein sample in its highest purity. Firstly, crude protein extract was subject to ion-exchange chromatography using the DEAE-cellulose column (20 × 1 cm). The prepared sample having approximately 3 mL of aqueous protein (using 3 mL loop) was poured into the DEAE-cellulose column. The column was first pre-equilibrated with potassium phosphate buffer (50 mM, pH 6.8) and then washed with equilibration buffer. The protein elutions were designed in the linear gradient of 0–500 mM NaCl and 0–100 mM potassium phosphate buffer keeping a 0.5 mL per min flow rate. The eluted fraction of protein obtained from the ion-exchange column was then applied to Superdex 200 pg column for gel filtration. The column was pre-equilibrated with a 50 mM phosphate buffer of pH 6.8. The protein was eluted in the equilibration buffer at a flow rate of 0.5 mL/min. The sample collection was performed at 4 °C. The active fractions were pooled, dialyzed against the 50 mM phosphate buffer of pH 6.80, and concentrated [44]. The fractions (3 mL each) were collected for SDS-PAGE. An SDS-PAGE of 12% separating gel and 4% stacking gel were prepared by the method of Laemmli, 1970 [45]. The previously pooled samples were prepared by adding 1% (*w*/*v*) SDS and then boiling for 5 min at 100 °C in Eppendorf tubes. To visualize the purified protein, gel electrophoresis was run in Tris-HCl buffer of pH 8.4 at 80–100 V for 3 hr. After electrophoresis, the protein bands on the gel were made detectable by staining with standard Coomassie Brilliant Blue. Samples showing the discrete band in SDS-PAGE were assayed for tyrosinase activity [44,46].

#### 3.2.4. Tyrosinase Activity Assay

The enzyme activity of tyrosinase was assessed by performing absorption spectroscopy where the rate of conversion of L-DOPA (substrate) to dopachrome (a red-colored oxidized product) was monitored. An aliquot containing purified tyrosinase enzyme was incubated for 5 min at 25 °C, L-DOPA solution (4 mg/mL) as substrate was added to the reaction mixture for measurement at 473 nm. After incubation for additional 5 min, the mixture was shaken again, and a second reading was measured for 10 min. The change in absorbance appeared to be proportional to enzyme concentration [47]. Zymography was also performed in order to measure the enzymatic activity of the purified enzyme. A similar protocol was followed to that explained by Wilkesman et al. [48]. 

#### 3.2.5. Tyrosinase Inhibition Kinetics and UV-Visible Spectral Measurements 

The anti-tyrosinase activity of santalol was measured using UV-Vis spectroscopy. The stock solution of santalol (1 mM) was prepared using phosphate buffer solution. Absorption measurements were carried out using a Jasco F-660 UV spectrometer in a 1.0 cm quartz cuvette at room temperature. The change in absorbance at 473 nm was measured by subsequently adding santalol but keeping protein concentration constant in each reading as described in various reports [49,50]. We also monitored overall spectral changes before and after addition of santalol in the purified tyrosinase enzyme at pH 6.8. Before every measurement, all the samples were incubated for 5–10 min at RT and the spectra were measured in the range of 240–340 nm [36].

#### 3.2.6. Intrinsic Fluorescence Binding Study 

The binding affinity of santalol towards purified protein tyrosinase was evaluated by examining changes in the fluorescent intensities [51,52]. These quenching experiments were performed on a spectrofluorometer (Model no. FP-6200). Slit widths of 10 nm and a quartz cuvette with 1 cm path length were used for both emission and excitation, and temperatures were set to 25 °C. Measurements were performed with 25 μM tyrosinase in 30 mM potassium phosphate buffer of pH 6.8, with excitation wavelength set at 280 nm and emission spectra recorded between 300 and 400 nm. Mathematical evaluation of fluorescence quenching data was performed using Stern-Volmer and modified SternVolmer plots [51]. 

#### 3.2.7. Isothermal Titration Calorimetry (ITC) 

To evaluate the thermodynamic parameters and binding interaction of tyrosinase with ligand molecules in the buffer solution, ITC was used. The isothermal titration calorimetry (ITC) is the excellent method to elucidate binding interaction and for that VP-ITC Calorimeter (MicroCal, 22 Industrial Drive East, Northampton, MA, USA) apparatus was utilized. We calibrated the instrument before carrying out the ITC experiment and also washed it with double-distilled water twice to make the sample cell free of any unwanted impurity. The experiments were carried out at 25 °Cat pH 6.8 using 25 mM phosphate buffer, and the calorimeter cell was injected with a 30 µM of the protein solution (tyrosinase). Degassing was performed for about 20–30 min. to ensure no air bubble is formed in the samples. This step plays important role to avoid the unwanted foam formation in the sample wall of ITC jacket. Therefore, standard procedure of degassing was employed using MicroCal system installed with ITC instrument, based on vacuum processing, which removes air trapped in form of small bubbles or foam in each sample of the protein and ligand. The ligand (santalol) of 900 µM was titrated in the cell, and each ligand solution was loaded with 10-microliter aliquots in each 260 s step through the syringe. The data was normalized against the results of titration of santalol into tyrosinase and was evaluated by the MicroCal Origin ITC software with a three-step sequential binding model as reported earlier [53,54,55], which could fit to the data to generate the change in binding enthalpy (Δ*H*), change in entropy (Δ*S*), and the association constant (*K*_a_). By these primary measurements, secondary parameter changes in Gibbs free energy (Δ*G*^o^) can be calculated by using the relation:(3)$$\Delta G{}^{\circ}=-RT\;\ln K_{a}=\Delta H-T\Delta S$$
where *R* is the gas constant and *T* is the absolute temperature.

The heat of dilution of the ligand in phosphate buffer was subtracted from the titration data. MicroCal Origin 8.0 was used to calculate the stoichiometry of association constant (*K*_a_), enthalpy change (Δ*H*), and binding (*n*).

#### 3.2.8. Molecular Docking Analysis

To obtain more comprehensive information about the inhibitory mechanism between santalol and tyrosinase, molecular docking studies were performed by using Autodock 4.0. The three-dimensional (3D) crystal structure of *Agaricus bisporus* tyrosinase (PPO3) was obtained from RCSB PDB databank PDB ID: 2Y9X (https://www.rcsb.org/structure/2Y9X/, accessed on 1 October 2022). InstaDock [37] was used in a molecular docking learning approach which prepares the receptor molecule pdb format into pdbqt with a simple click [33105480; 34293449]. PPO3 (tyrosinase) includes two copper atoms that participate directly involved in the numerous catalytic activities. The target ligands were docked on a predefined catalytic site using PyMOL and AutoDock Vina to determine the 3D grids of the target protein PPO3 [30521996; 19499576; 34469971]. The compound’s structure was derived from PubChem database (https://pubchem.ncbi.nlm.nih.gov/, accessed on 1 October 2022) in SDF format for ligand preparation, where the energy minimization and PDBQT file conversion was done via PyRx 0.8 [25618350]. Assortment of the binding affinities scores further switched on the studies of the receptor–ligand interaction which interprets the interacted residues in the catalytic sites. The conceivable dock conformations of 2D ligand–receptor interactions were presented by using Discovery Studio [20401516; 33592201]. Further, binding analysis by using the visualization approach was carried out to understand the binding pattern of the ligands with a receptor. Then, further evaluation was carried out based on interactions to avoid false positives and select examples highly interactive with the binding pocket of PPO3.

## 4. Conclusions

Tyrosinase is the major enzyme responsible for the synthesis of melanin pigment, but the hyper-pigmentation associated with the enhanced activity of tyrosinase is highly undesirable. It promotes unwanted food browning and dermatological disorders in humans. Thus, screening and elucidation of the potential inhibitor for this enzyme serve as good applications in commercial domains such as agriculture, medicine, the cosmetics industry, and other pharmacological sectors. Despite the presence of a wide array of reported tyrosinase inhibitors, very few of them are non-toxic and effective at the same time. In this study, we propose a safe to use phytochemical called santalol derived from sandalwood with anti-tyrosinase activity. A plausible kinetic mechanism analyzed by UV-visible spectroscopy is presented where santalol mediates competitive inhibition as it fits into the catalytic pocket of the enzyme and alters its structure, which hinders the binding of the original substrate L-tyrosine. Furthermore, the fluorescence quenching study and isothermal titration calorimetry results also suggested that there is a very strong binding between tyrosinase and the test compound santalol, in accordance with the molecular docking study. Our results indicate that santalol may serve as a novel as well as natural anti-tyrosinase agent, although the clinical and industrial trials are yet to be elucidated. Santalol has been used since ancient times for beautification purposes, but a detailed mechanism of action of its inhibition has not been unveiled; therefore, this is the first study which provides a possible mechanism for santalol’s action on tyrosinase activity as well as the effect of the santalol compound on structural integrity. However, the findings we attained here in this study need further investigation in pigment cell assays or in animal models and in clinical studies.

## Figures and Tables

**Figure 1 molecules-27-08915-f001:**
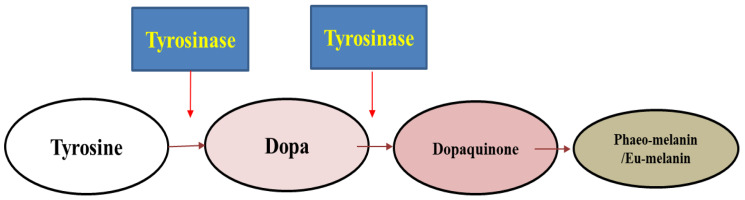
Conversion of tyrosine into melanin in presence of tyrosinase enzyme (tyrosine to melanin conversion pathway) [19].

**Figure 2 molecules-27-08915-f002:**
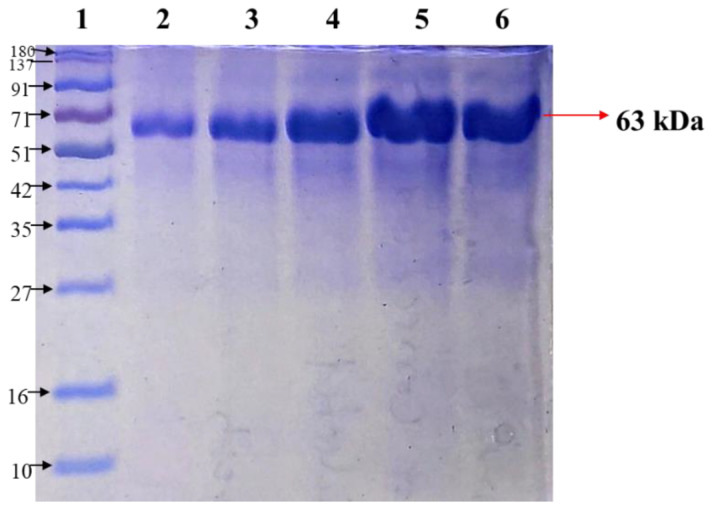
SDS-PAGE profile indicating 63 kDa purified form of tyrosinase, with lane 1 showing protein marker and lanes 2–6 depicting the purified tyrosinase loaded in gradient (lanes 5, 6 represent highest concentration of protein).

**Figure 3 molecules-27-08915-f003:**
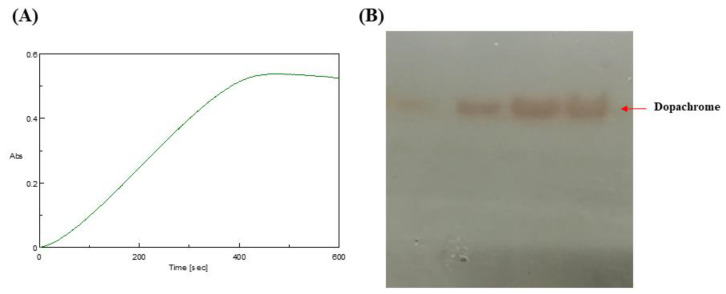
(**A**) Change in absorbance was monitored at 473 nm (dopachrome λ_max_ = 473). (**B**) Dopachrome was formed upon addition of tyrosine as substrate in an enzyme mixture proving the active form of tyrosinase in reaction mixture. Native-PAGE gel was incubated in 5 mM L-tyrosine (substrate) at 37 °C, and bands of light red-colored pigment dopachrome (shown right side) were appeared after a few minutes of incubation, which eventually leads to production of melanin dark brown color (shown in Figure 3 right side).

**Figure 4 molecules-27-08915-f004:**
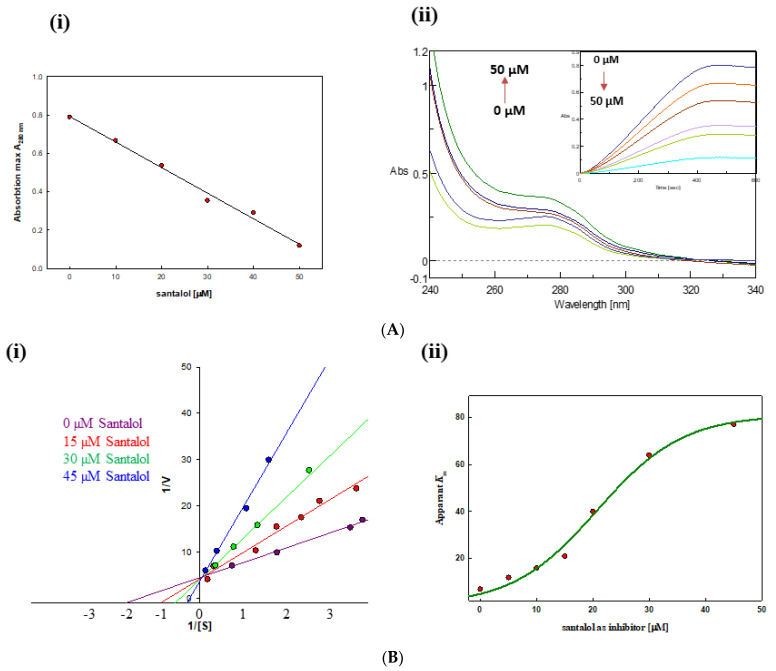
(**A**) (**i**) Effect of various concentrations of compound santalol as tyrosinase inhibitor on absorbance maxima A_280 nm_ values. (**ii**) Effects of varying concentrations of santalol on tyrosinase tertiary structure monitors using near-UV visible spectroscopic scan, with inset figure showing progressive tyrosinase inhibition with increase in santalol concentration (µMolml^−1^). (**B**) (**i**) Lineweaver–Burk plot in presence of various concentrations of santalol, with right-hand side (**ii**) depicting increase in *K*_m_ values were also observed in dose-dependent manner.

**Figure 5 molecules-27-08915-f005:**
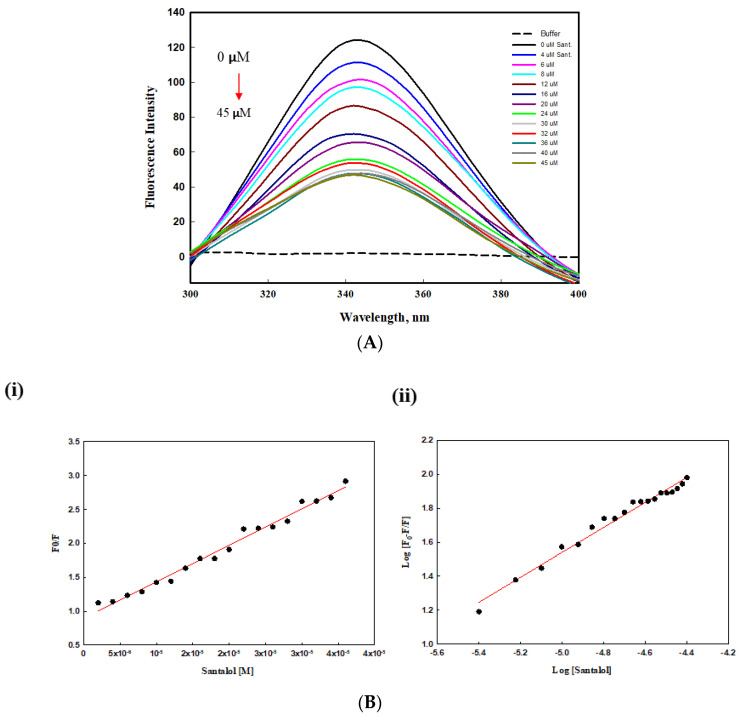
(**A**) Intrinsic fluorescence spectra of tyrosinase in the presence santalol at various concentrations, which range from 0 to 45 µM for top to bottom curves, respectively. (**B**) Stern-Volmer (**i**) and modified Stern-Volmer plots (**ii**) for the fluorescence quenching of tyrosinase at 25 °C by santalol (tyrosinase–santalol).

**Figure 6 molecules-27-08915-f006:**
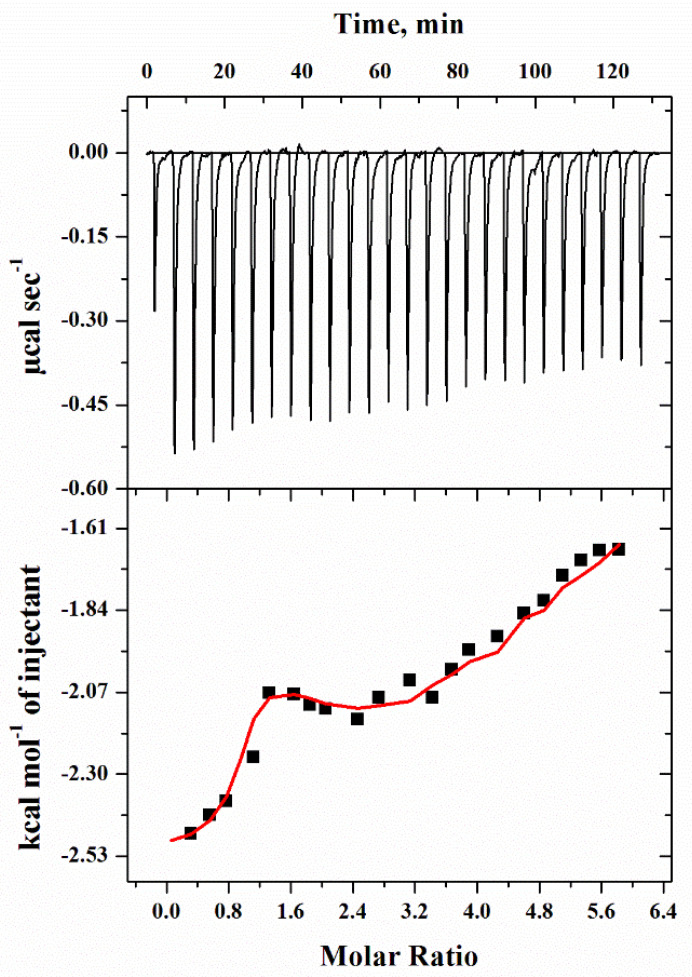
Isothermal titration calorimetry (ITC) profile of tyrosinase-santalol binding: calorimetric responses owing to consecutive injections of santalol in the ample cell with tyrosinase are depicted in upper section, while the lower section depicts integrated heats of interactions as the function of the [santalol]/[tyrosinase] molar ration.

**Figure 7 molecules-27-08915-f007:**
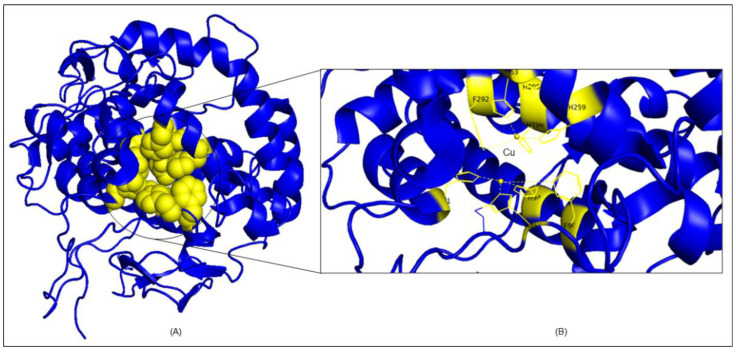
Ligand binding pocket studies. (**A**) Cartoon representation of PPO3 tyrosinase PDB ID: 2Y9X (blue) shows the catalytic sites in sphere shape (yellow). (**B**) Nearby copper (Cu) atom residues are involved in various catalytic activities.

**Figure 8 molecules-27-08915-f008:**
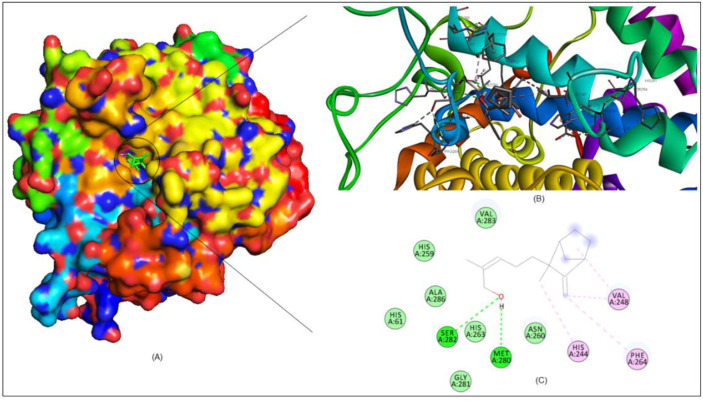
Molecular interaction studies showing the santalol–PPO3 tyrosinase complex. (**A**) The surface model of santalol-PPO3 tyrosinase complex was in active pocket. (**B**) Cartoon representation of tyrosinase protein complex with santalol showing the interacted residues in expanded form. (**C**) The 2D interaction analysis of complex files was done by BIOVIA here, and the different color showing bonds are light green—carbon hydrogen bond; green—hydrogen bond; and purple—Pi alkyl bond.

**Table 1 molecules-27-08915-t001:** Purification summary of tyrosinase.

S. No.	Purification Steps	Fraction Volume (mL)	Protein Conc. (mg/mL) ^b^	Total Amt. of Protein (mg)	Total Activity (Units)	Specific Activity (Unit/mg)	Yields (%)	Purification Fold
**1**	**Crude extract ^a^**	310.0	1.250	387.5	4121	10.6	100	1
**2**	**Ammonium sulphate precipitation 30% cut-off**	120	0.853	102.36	3172	30.9	26.42	2.91
**3**	**Ion-exchange chromatography, DEAE-Sepharose column**	65	0.552	35.88	2627.5	73.25	9.25	7.07
**4**	**Gel filtration chromatography, using Superdex 200 column**	45	0.15	7	2256.2	322.3	1.8	32.14

**^a^** from 200 g of wet weight of mushroom cap edible portion. **^b^** Protein concentration determined by Bradford assay [35] using BSA as a standard protein.

**Table 2 molecules-27-08915-t002:** Calorimetric binding parameters obtained by analysis of ITC measurements on interaction of santalol with purified tyrosinase at 25 °C) and pH 6.8.

Thermodynamic Parameters (Units)	Step 1	Step 2	Step 3
** *K* _a_ ** **(M^−1^)**	7.89 × 10^5^ ± 8.5 × 10^4^	1.32 × 10^4^ ± 1.7 × 10^3^	2.49 × 10^3^ ± 2.8 × 10^2^
**∆*H*** **(cal mol^−1^)**	−2.595 × 10^3^	−6.497 × 10^3^	−4.16 × 10^4^
**∆*S*** **(cal mol^−1^deg^−1^)**	18.3	−2.94	−124
**∆*G*°** **(cal mol^−1^)**	−8.048 × 10^3^ ± 90.0	−5.62 × 10^3^ ± 7.87 × 10^2^	4.16 × 10^3^ ± 90.0

## Data Availability

All data generated or analyzed during this study are included in this manuscript and Appendix A attached to this article.

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
