# Peer review of "Elucidating the Role of Santalol as a Potent Inhibitor of Tyrosinase: In Vitro and In Silico Approaches"

_molecules, 2022, doi:10.3390/molecules27248915_

Round 1

Reviewer 1 Report

This research article focuses on the potential application of santalol in the inhibition of the enzyme tyrosinase, which is actively involved in the biosynthesis of melanin pigment. Over-production of melanin causes undesirable pigmentation in humans as well as other organisms that significantly downgrade their aesthetic value. The study is designed to explain the purification of tyrosinase from the mushroom Agaricus bisporus, followed by activity assay and enzyme kinetic to provide insight into the santalol modulated tyrosinase inhibition in a dose-dependent manner. The multi-spectroscopic techniques like UV-vis, fluorescence, and isothermal calorimetry are employed to deduce the efficiency of santalol as a potential candidate against tyrosinase enzyme activity. The authors showed that santalol inhibits tyrosinase, which is already known to some extent, but this investigation provides a mechanism for santalol action, so it is a novel and important finding. The work is elaborate, and the results look impressive. The manuscript is well written with a good application, but many things are missing at several points. This paper may be suitable for publication following revision. The authors should answer and resolve the following queries.

1.     A suitable reference should be given in Figure 1.

2.     Material and Methods: what was the concentration of Santalol in the study? 

3.     Provide the detail about purity santalol.

4.     Figure 4A: Legend may be corrected. It is not far UV-visible spectroscopic scan.

5.     Figure 5A: The figure shows a santalol concentration of up to 45 µM while the legend says up to 40 µM. Kindly remove the discrepancy.

6.     Figure 5B: what does [comp] means on the x-axis?

7.     ITC is a very sensitive technique. How were the calibration and baseline performed?

8.     How was the degassing done for ITC experiments?

9.     How do the interaction studies done by ITC validate with molecular docking studies?

10.  Reference may be given for InstaDock.

11.  Grammatical mistakes may be removed from the manuscript. Authors are suggested to check for grammatical and typos mistakes throughout the manuscript.

12.  Line no. 99: santalol may be written with small s.

13.   “Similar Protocol was”…why P capital?

14.  … “Michaelis-Menten equation”….should be written as Michaelis-Menten equation.

15.  The literature review should be more current.

Author Response

Point by Point Response to the Reviewers

Manuscript title: Elucidating the role of santalol as a potent inhibitor of tyrosinase: In vitro and In silico approaches

Manuscript ID: molecules-2042120

Journal: (Molecules) MDPI

First of all, we would like to thank the Editor for considering this work for revision. Moreover, we are thankful to the Reviewers for providing valuable comments to improve the revised manuscript. As per the suggestions of Reviewers, typographical and grammatical errors have been corrected and overall quality of the paper is improved in terms of science. Appropriate changes have been made in the revised manuscript as well as in the figures.

Reviewer: 1

This research article focuses on the potential application of santalol in the inhibition of the enzyme tyrosinase, which is actively involved in the biosynthesis of melanin pigment. Over-production of melanin causes undesirable pigmentation in humans as well as other organisms that significantly downgrade their aesthetic value. The study is designed to explain the purification of tyrosinase from the mushroom Agaricus bisporus, followed by activity assay and enzyme kinetic to provide insight into the santalol modulated tyrosinase inhibition in a dose-dependent manner. The multi-spectroscopic techniques like UV-vis, fluorescence, and isothermal calorimetry are employed to deduce the efficiency of santalol as a potential candidate against tyrosinase enzyme activity. The authors showed that santalol inhibits tyrosinase, which is already known to some extent, but this investigation provides a mechanism for santalol action, so it is a novel and important finding. The work is elaborate, and the results look impressive. The manuscript is well written with a good application, but many things are missing at several points. This paper may be suitable for publication following revision. The authors should answer and resolve the following queries.

We are grateful to the esteemed reviewer for critical evaluation of the manuscript. Their comments have indeed important to raise the quality of the manuscript. We are hereby providing a point-to-point response to reviewer’s comments in addition to revision of the manuscript as per the suggestion of the reviewers. We are highly thankful to the respected Reviewer’s appreciation for the work.

Comment 1: A suitable reference should be given in Figure 1.

Response: Suitable current reference for Figure 1 has been incorporated in the revised manuscript (in Figure 1 legend).

Comment 2: Material and Methods: what was the concentration of Santalol in the study? 

Response: Concentration of santalol used in this study was 1 mM (stock) from which we have used santalol in working concentrations ranges from 0-50 µM using dilution method. The changes has been made in the revised manuscript.

Comment 3: Provide the detail about purity santalol.

Response: Santalol used in this study as a natural inhibitor was purchased from sigma Aldrich (about 93% pure). The certificate of analysis (COA) link is attached below.

https://www.sigmaaldrich.com/certificates/Graphics/COfAInfo/fluka/pdf/PDF689559.pdf

Comment 4:     Figure 4A: Legend may be corrected. It is not far UV-visible spectroscopic scan. 

Response:  From Figure 4A, typographical error has been removed from the legend and the text and updated in revised manuscript.

Comment 5:     Figure 5A: The figure shows a santalol concentration of up to 45 µM while the legend says up to 40 µM. Kindly remove the discrepancy.

Response: In figure 5A, we have done fluorescence quenching experiments, where we have titrated the purified tyrosinase with compound (santalol as ligand).  The discrepancy has been removed by correcting the ligand concentration in the legend in the revised manuscript.

Comment 6:    Figure 5B: what does [comp] means on the x-axis?

Response: In Figure 5B, x-axis legend “[comp]” has been replaced by “[santalol]” in the revised manuscript.

Comment 7:    ITC is a very sensitive technique. How were the calibration and baseline performed?

Response: Thanks!! ITC is amongst the most sensitive technique employed to study protein-ligand interactions and have an insight into the thermodynamic parameters of the system. Throughout an ITC experiment, a small constant amount of power (equal to the Reference Power entered) is continuously supplied to the offset heater of the reference cell. This causes the DP feedback system to become positive to supply compensating power to the sample cell that will equilibrate the temperatures. The Initial Delay refers to the time, in seconds, after the instrument has started a run and before the first injection. The standard parameter is 60 seconds; this is necessary to establish a baseline prior to the first injection. All these were taken care of before starting the experiment. Description is added in the revised manuscript.

Comment 8:     How was the degassing done for ITC experiments?

Response: Degassing of cell and syringe samples is a vital step to ensure their bubble- free loading. A ThermoVac sample degassing and thermostat station is provided with each instrument. The ThermoVac's preset vacuum time of ~ 5 minutes is adequate to degas a sample that is being stirred, but a much longer time is required without stirring. The standard practice is to set the ThermoVac temperature to 1 to 5 degrees below the intended experimental temperature while degassing. Therefore, standard procedure of degassing was employed using MicroCal system installed with ITC instrument, based on vacuum processing which removes air trapped in form of small bubbles or foam in each sample of the protein and ligand. We carried out degassing for 30 minutes without stirring. Description is added in the revised manuscript.

Comment 9:     How do the interaction studies done by ITC validate with molecular docking studies?

Response: ITC is amongst the most sensitive technique that deciphers the binding affinity and delineates the thermodynamic parameters of the system. Molecular docking reveals the critical residues involved in the binding process. The experiential binding free energy for the santalol-PPO3 tyrosinase complex was -5.802 kcal mol-1 (-24.283 k J mol-1) and the standard Kojic acid complex was -5.408 mol-1 (-22.608 k J mol-1). ITC demonstrates that santalol binds to tyrosinase with significant binding affinity which is also validated by molecular docking.

Comment 10:  Reference may be given for InstaDock.

Response: Reference, where we have discussed about InstaDock has been added in revised manuscript.

Comment 11: Grammatical mistakes may be removed from the manuscript. Authors are suggested to check for grammatical and typos mistakes throughout the manuscript.

Response: All typographical and grammatical mistakes have been removed from the revised Manuscript.

Comment 12:  Line no. 99: santalol may be written with small s.

Response: In whole manuscript “Santalolis replaced by “santalol”.

Comment 13: … “Similar Protocol was”…why P capital?

Response: Typographical error has been removed from revised manuscript.

Comment 14:  … “Michaelis-Menten equation”….should be written as Michaelis-Menten equation.

Response: We have made the required changes and replaced “Michaelis-menten” to “Michaelis-Menten” in the revised manuscript.

Comment 15:  The literature review should be more current.

Response: As per suggestion of reviewer suitable recent reference has been added in introduction after a profound review of the literature again and few more details regarding the experiment have been added in the revised manuscript.

Reviewer 2 Report

The article titled "Elucidating the role of santalol as a potent inhibitor of tyrosinase: In vitro and In silico approaches" is very interesting. However, some parts should be improved before accepting in a highly renowned journal such as Molecules. 

In line 80 full name of the species should be written.

In my opinion, in the whole text temperature should be written in the same units (not somewhere 298 K, somewhere 37 °C, somewhere 37 degrees).

In line 249 is said: „Protein concentration determined by Bradford assay [55] using BSA as a standard protein.“ However, there are only 51 references in the reference list at the end of the paper. 

In lines 280 ana 284 is written „Michaelis-menten“ instead of Michaelis-Menten.

In my opinion, in all assays santalol should be compared with kojic acid, as a standard inhibitor, the same as it was done in part about Molecular docking analysis.   

In Figure 4. A) i and Figure 4. B) ii is written uM instead of µM. 

In line 324 is written 0-40 µM, while in Figure is shown from 0-45 µM. 

In lines 331 and 332 is written „kq = Bimolecular quenching constant τ0 = Lifetime of the fluorophore in the absence of the quencher”, However, these values are not used in the equation.

 Was toxicity evaluation of currently used inhibitors performed with machine learning methods using the pkCSM server?

Author Response

Point by Point Response to the Reviewers

Manuscript title: Elucidating the role of santalol as a potent inhibitor of tyrosinase: In vitro and In silico approaches

Manuscript ID: molecules-2042120

Journal: MDPI (Molecules)

First of all, we would like to thank the Editor for considering this work for revision. Moreover, we are thankful to the Reviewers for providing valuable comments to improve the revised manuscript. As per the suggestions of Reviewers, typographical and grammatical errors have been corrected. Appropriate changes have been made in revised manuscript as well as in figures.

Reviewer: 2

The article titled "Elucidating the role of santalol as a potent inhibitor of tyrosinase: In vitro and In silico approaches" is very interesting. However, some parts should be improved before accepting in a highly renowned journal such as Molecules. 

Response: We are thankful to the Reviewer for commenting and finding the current work “very interesting”. We are highly thankful to the respected Reviewer’s appreciation for the work. Moreover, we are grateful to the esteemed reviewer for critical evaluation of the manuscript. Their comments have indeed important to raise the quality of the manuscript. We are hereby providing a point-to-point response to reviewer’s comments in addition to revision of the manuscript as per the suggestion of the reviewers.

In the revised manuscript, the line number have been changed, therefore we have mentioned the new line number at the end of every response.

Comment 1: In line 80 full name of the species should be written.

Response: In line 80 full name of the species Agaricus bisporus (mushroom) has been written in revised manuscript.

Comment 2: In my opinion, in the whole text temperature should be written in the same units (not somewhere 298 K, somewhere 37 °C, somewhere 37 degrees).

Response: As per suggestion of reviewer, we have incorporated the required changes in the revised manuscript. We have changed the temperature unit K/ degree to “°C” in the whole manuscript.

Comment 3: In line 249 is said: ,,Protein concentration determined by Bradford assay [55] using BSA as a standard protein.“ However, there are only 51 references in the reference list at the end of the paper. 

Response: In Line 249, we have incorporated the correct reference and removed the typographical errors, as per the suggestion of the reviewer.

Comment 4: In lines 280 and 284 is written „Michaelis-menten“ instead of Michaelis-Menten.

Response: We have made the required changes and replaced “Michaelis-menten” to “Michaelis-Menten” in the revised manuscript.

Comment 5: In my opinion, in all assays santalol should be compared with kojic acid, as a standard inhibitor, the same as it was done in part about Molecular docking analysis.   

Response:  We have considered kojic acid as the standard inhibitor of tyrosinase in this study. The maximum inhibitory concentration of santalol was compared with the maximum inhibitory concentration of kojic acid. The required data of the comparative colorimetric assay is presented in the revised manuscript as supplementary data (Figure S4).

Comment 6: In Figure 4. A) i and Figure 4. B) ii is written uM instead of µM. 

Response: We have made the suggested changes in Figure 4A, i and ii, “uM” is replaced byµM”.

Comment 7: In line 324 is written 0-40 µM, while in Figure is shown from 0-45 µM. 

Response: In line 324, typographical error has been removed and corrections has been made in revised manuscript (Figure 5 A).

Comment 8: In lines 331 and 332 is written„ kq = Bimolecular quenching constant τ0 = Lifetime of the fluorophore in the absence of the quencher”, However, these values are not used in the equation.

Response: As suggested by reviewer, The static quenching interaction is calculated by equation (3), where the aforementioned variables are not incorporated into the equation. Hence, the typographical errors in Line 331 and 332 were removed and correction has been done in revised manuscript.

Comment 9: Was toxicity evaluation of currently used inhibitors performed with machine learning methods using the pkCSM server?

Response:  Yes, we have performed the same for toxicity evaluation of other currently used inhibitors. Using the pkCSM server, which is a machine learning platform to predict small molecule's pharmacokinetics properties, which relies on distance/pharmacophore pattern encoded as graph-based signatures. We have assessed the drug toxicity of the santalol versus kojic acid/hydroxyquinone/arbutin etc. (standard inhibitor of tyrosinase) [1] on various parameters viz. AMES toxicity; hERG 1, 2 inhibitors; hepatotoxicity etc. Eventually, we have found similar toxicity index of santalol for these parameters when compared to its counterparts as a tyrosinase inhibitor.

References:

  1. Zolghadri, S., et al., A comprehensive review on tyrosinase inhibitors. Journal of Enzyme Inhibition and Medicinal Chemistry, 2019. 34(1): p. 279-309.

Round 2

Reviewer 2 Report

The authors improved their paper and replied to all questions.